# Exploring the Interconnection between Metabolic Dysfunction and Gut Microbiome Dysbiosis in Osteoarthritis: A Narrative Review

**DOI:** 10.3390/biomedicines12102182

**Published:** 2024-09-25

**Authors:** Hui Li, Jihan Wang, Linjie Hao, Guilin Huang

**Affiliations:** 1Department of Joint Surgery, Honghui Hospital, Xi’an Jiaotong University, Xi’an 710054, China; 2Institute of Medical Research, Northwestern Polytechnical University, Xi’an 710072, China

**Keywords:** osteoarthritis, metabolic dysfunction, gut microbiome, joint inflammation, cartilage degradation

## Abstract

Osteoarthritis (OA) is a prevalent joint disorder and the most common form of arthritis, affecting approximately 500 million people worldwide, or about 7% of the global population. Its pathogenesis involves a complex interplay between metabolic dysfunction and gut microbiome (GM) alterations. This review explores the relationship between metabolic disorders—such as obesity, diabetes, and dyslipidemia—and OA, highlighting their shared risk factors, including aging, sedentary lifestyle, and dietary habits. We further explore the role of GM dysbiosis in OA, elucidating how systemic inflammation, oxidative stress, and immune dysregulation driven by metabolic dysfunction and altered microbial metabolites contribute to OA progression. Additionally, the concept of “leaky gut syndrome” is discussed, illustrating how compromised gut barrier function exacerbates systemic and local joint inflammation. Therapeutic strategies targeting metabolic dysfunction and GM composition, including lifestyle interventions, pharmacological and non-pharmacological factors, and microbiota-targeted therapies, are reviewed for their potential to mitigate OA progression. Future research directions emphasize the importance of identifying novel biomarkers for OA risk and treatment response, adopting personalized treatment approaches, and integrating multiomics data to enhance our understanding of the metabolic–GM–OA connection and advance precision medicine in OA management.

## 1. Introduction

Osteoarthritis (OA), the most prevalent form of arthritis, impacts approximately 500 million individuals globally, accounting for about 7% of the world’s population. Various epidemiological studies have shed light on the burden of OA across different populations, providing insights into its prevalence, incidence, and associated risk factors [1,2]. OA is a complex, multifaceted disease that affects all joint tissues. It involves subchondral bone remodeling and meniscal degeneration, as well as inflammation and fibrosis of both the infrapatellar fat pad (IFP) and synovial membrane [3,4]. This comprehensive understanding of OA as a whole-joint disease underscores the importance of therapeutic approaches that target all affected tissues, rather than focusing solely on individual components [5]. Concurrently, there has been a notable surge in the incidence of metabolic disorders such as obesity, diabetes, and dyslipidemia [6]. These disorders are characterized by disruptions in metabolic homeostasis, leading to adverse health outcomes and contributing to the development and progression of various chronic diseases, including OA [6,7]. Remarkably, there exists a significant overlap in the risk factors and comorbidities between OA and metabolic dysfunction [8,9]. Shared risk factors such as aging, sedentary lifestyle, and dietary habits contribute to the co-occurrence of these conditions, exacerbating their impact on affected individuals’ quality of life [10,11].

The gut microbiome (GM), comprising trillions of microorganisms inhabiting the gastrointestinal tract, plays a crucial role in maintaining host health and homeostasis [12,13]. Composed primarily of bacteria, fungi, viruses, and archaea, the GM exhibits a remarkable diversity that varies across individuals and is influenced by multiple factors such as diet, genetics, and environmental exposure [14]. The GM actively interacts with the host physiology through different mechanisms, including the fermentation of dietary fibers, the production of metabolites, the modulation of immune responses, and the regulation of metabolic processes. These interactions are essential for nutrient absorption, energy metabolism, and overall gut health, highlighting the symbiotic relationship between the host and its microbial inhabitants [15,16]. Furthermore, the GM exerts profound effects on immune regulation and metabolic homeostasis. Through the production of immunomodulatory factors and the maintenance of intestinal barrier integrity, gut microbes play a crucial role in shaping the host’s immune system and defending against pathogens [17,18]. Additionally, alterations in GM composition have been linked to metabolic disorders such as obesity and diabetes, underscoring the intricate interplay between GM and metabolic health [19,20].

Emerging evidence suggests that metabolic dysfunction and OA share common pathophysiological mechanisms, implicating systemic inflammation, oxidative stress, and insulin resistance in the development and progression of both conditions [8,9,21,22]. The dysregulation of metabolic pathways, such as adipokine signaling and lipid metabolism, may contribute to the onset of OA and exacerbate its symptoms. Moreover, recent studies have highlighted the role of GM dysbiosis in the pathogenesis of metabolic disorders and OA. Dysfunctional GM composition and altered microbial metabolite profiles have been associated with insulin resistance, obesity, and dyslipidemia, providing a potential link between gut dysbiosis and metabolic dysfunction-driven OA [2,23]. Given the interconnectedness of metabolic dysfunction, GM alterations, and OA pathology, targeting these interrelated pathways holds promise for therapeutic advancements in OA management [24,25]. Strategies aimed at modulating the GM composition through dietary interventions, probiotics, or fecal microbiota transplantation (FMT) may offer novel therapeutic avenues for alleviating OA symptoms and slowing disease progression [26,27].

The aim of this review is to systematically analyze how GM dysbiosis influences the development and progression of OA through metabolic pathways. We explore the association between metabolic syndrome and OA, with a specific focus on the impact of microbial metabolites and signaling molecules on the pathophysiology of OA. In addition to summarizing the existing literature, this review introduces new insights into the underexplored metabolic signaling pathways involved in the gut–joint axis. By doing so, we aim to provide a foundation for future therapeutic strategies targeting the GM for OA treatment, contributing novel perspectives to this emerging research area.

## 2. Metabolic Dysfunction and OA

Metabolic dysfunction, encompassing conditions such as metabolic syndrome, insulin resistance, and dyslipidemia, has emerged as a significant contributor to the pathogenesis and progression of OA. Understanding the intricate connections between metabolic derangements and OA provides crucial insights into potential therapeutic targets and management strategies for this debilitating joint disorder [8,22]. Figure 1 visualizes the interconnections between metabolic dysfunction and OA.

### 2.1. Metabolic Syndrome and Its Association with OA

Metabolic syndrome, characterized by a cluster of metabolic abnormalities including central obesity, insulin resistance, dyslipidemia, and hypertension, has been closely linked to the development and severity of OA [28,29]. Epidemiological studies have consistently demonstrated a positive association between metabolic syndrome and OA, particularly in weight-bearing joints such as the knees and hips. The chronic low-grade inflammation and dysregulated lipid metabolism characteristic of metabolic syndrome contribute to joint tissue damage and cartilage degradation, ultimately accelerating OA progression.

### 2.2. Adipokines, Inflammation, and OA Pathogenesis

Adipose tissue, once considered merely a storage depot for fat, is now recognized as an active endocrine organ capable of secreting a myriad of bioactive molecules known as adipokines. The dysregulation of adipokine signaling, characterized by elevated levels of proinflammatory adipokines such as leptin and reduced levels of anti-inflammatory adiponectin, plays a pivotal role in the pathogenesis of OA. Adipokines exert direct effects on joint tissues, promoting inflammation, cartilage degradation, and osteophyte formation. Moreover, adipokines contribute to systemic inflammation and insulin resistance, further exacerbating metabolic dysfunction and perpetuating OA pathology [28,30].

### 2.3. Role of the Infrapatellar Fat Pad in OA Inflammation

The IFP plays a significant role in joint inflammation in OA [31]. This tissue is a source of various cytokines and adipokines, including proinflammatory molecules such as leptin and interleukin-6 (IL-6) [30,32]. These secreted factors contribute to local joint inflammation, exacerbating synovial inflammation and cartilage degradation. The IFP’s interaction with the synovium and its role in producing inflammatory mediators make it a key target for future therapeutic interventions aimed at reducing OA progression [33,34].

### 2.4. Impact of Insulin Resistance and Dyslipidemia on Joint Health

Insulin resistance, a hallmark feature of type 2 diabetes mellitus (T2DM) and metabolic syndrome, has emerged as a significant contributor to OA pathogenesis [35]. Insulin resistance disrupts normal chondrocyte function and extracellular matrix (ECM) metabolism, leading to cartilage degeneration and impaired joint homeostasis [8,36]. Additionally, hyperinsulinemia promotes the synthesis of proinflammatory cytokines and matrix metalloproteinases (MMPs), further fueling the inflammatory cascade within the joint microenvironment. Synovitis, often an early sign of OA, involves fibroblast-like synoviocytes (FLSs) that contribute to OA by secreting inflammatory cytokines like ILs, TNFs, MMPs, and ADAM metallopeptidase with thrombospondin type (ADAMTS) proteases, which degrade cartilage ECM. Hyperglycemia and hyperinsulinemia can trigger proinflammatory cytokines, including IL-6, TNF-α, and IL-1β, which promote inflammation, disrupt chondrocyte function, and impair ECM metabolism, potentially leading to joint damage in diabetes-related OA [37]. For instance, advanced glycation end products (AGEs), which result from proteins or lipids becoming glycated due to prolonged sugar exposure, tend to accumulate in aged cartilage tissue [38]. Moreover, extended insulin therapy, often essential for diabetes management, could potentially overburden tissues like cartilage. This is evidenced by the fact that joint damage tends to be more severe in individuals with diabetes [39,40]. Given that articular cartilage has a low rate of cell turnover [41], researchers propose that autophagy may play a significant role in modulating the impacts of hyperglycemia and hyperinsulinemia on joint health [42]. Furthermore, dyslipidemia, characterized by elevated levels of circulating lipids and lipoproteins, has been implicated in OA development and progression [43]. Oxidized lipids and cholesterol crystals accumulate in joint tissues, triggering inflammatory responses and promoting cartilage degradation. Studies have shown that dyslipidemia, particularly via oxidized low-density lipoprotein (LDL), disrupts normal autophagy by inhibiting the activity of transcription factor EB (TFEB), resulting in decreased autophagic flux and increased necroptosis in chondrocytes. This disruption accelerates cartilage degradation and promotes inflammation, positioning oxidized LDL as a key factor in cartilage damage associated with OA [44]. Similarly, research has shown that oxidative stress and lipid peroxidation play a significant role in the pathogenesis of rheumatoid arthritis (RA), leading to the production of harmful lipid hydroperoxides and reactive lipid species that trigger inflammation and contribute to cartilage degradation. Alterations in lipid metabolism, particularly in fatty acids, phosphatidylcholine, and phosphatidylethanolamine, further exacerbate inflammation, highlighting these lipids as potential biomarkers and therapeutic targets for RA [45].

## 3. The GM and OA

### 3.1. Gut Dysbiosis as a Contributing Factor to OA Development

The composition of the GM is profoundly influenced by various factors, including diet, host genetics, and environmental exposures. Perturbations in GM composition, commonly observed in metabolic disorders such as obesity and diabetes, have been implicated in OA pathogenesis [6,46]. Studies have revealed alterations in GM composition in individuals with OA compared to healthy controls, suggesting a potential link between gut dysbiosis and OA susceptibility. Moreover, shared risk factors and comorbidities between metabolic disorders and OA may further exacerbate alterations in GM composition, highlighting the interconnectedness of these conditions. To elucidate the intricate relationship between the GM and OA, Table 1 presents related research on this topic.

### 3.2. Mechanisms Underlying the Influence of GM on OA Pathophysiology

Gut dysbiosis represents a potential contributing factor to OA development and progression through multiple interconnected pathways, as illustrated in Figure 2. By elucidating the mechanisms underlying the influence of the GM on OA pathophysiology, researchers can identify novel therapeutic targets and develop personalized interventions for this prevalent joint disorder.

The mechanisms underlying the influence of gut dysbiosis on OA pathophysiology are multifaceted and involve intricate interactions between the GM, the host immune system, and joint tissues. Firstly, gut dysbiosis may contribute to OA development by promoting systemic inflammation. GM dysbiosis can trigger the release of proinflammatory cytokines and chemokines, leading to a state of chronic low-grade inflammation. Low-grade inflammation is characterized by a chronic, persistent inflammatory state that affects both the entire body (systemically) and specific tissues such as joints [74,75]. This inflammation often originates from metabolic disorders, including obesity and GM imbalances, triggering immune responses that contribute to the progression of OA [76]. Unlike acute inflammation, low-grade inflammation is subtle but continuous, leading to prolonged activation of immune cells, particularly macrophages, neutrophils, and fibroblasts, which release proinflammatory cytokines and chemokines. Among the key mediators are TNF-α, IL-1β, IL-6, and monocyte chemoattractant protein-1 (MCP-1). TNF-α plays a crucial role by inducing the production of enzymes like MMPs, which degrade cartilage ECM components while also promoting synovial inflammation [77,78]. IL-1β enhances the catabolic breakdown of cartilage by upregulating MMPs and ADAMTS, which further degrade collagen and proteoglycans, and it also suppresses anabolic cartilage repair processes [79]. IL-6 contributes to both systemic and local joint inflammation by promoting immune cell recruitment, increasing MMP production, and activating osteoclasts, thereby affecting both cartilage and bone integrity [77]. MCP-1 recruits monocytes and macrophages to inflamed joints, exacerbating synovial and cartilage destruction [80]. Additionally, adipokines such as leptin and adiponectin, produced by adipose tissue, link metabolic dysfunction to joint inflammation, with leptin enhancing the production of TNF-α and IL-6, further intensifying the inflammatory response in OA [81]. Collectively, these cytokines and chemokines create a feedback loop of inflammation, cartilage degradation, and impaired repair, leading to progressive joint damage, loss of function, and pain. Therefore, the low-grade systemic inflammation associated with metabolic dysfunction and GM dysbiosis is a key driver of both systemic and joint-specific inflammation in OA, highlighting the central role of these inflammatory mediators in cartilage degradation [82].

Additionally, bacterial metabolites and signaling molecules play a crucial role in influencing metabolic signaling pathways that contribute to the pathogenesis of OA. Key bacterial metabolites, particularly short-chain fatty acids (SCFAs) such as butyrate, acetate, and propionate, are produced through the fermentation of dietary fibers by GM [83,84]. These SCFAs interact with G-protein-coupled receptors (GPCRs), such as GPR41 and GPR43, which are expressed in adipose tissue, immune cells, and the intestinal epithelium [85]. The activation of these receptors regulates immune responses, inflammation, and energy metabolism, contributing to systemic metabolic balance. However, GM dysbiosis disrupts SCFA production and impairs AMP-activated protein kinase (AMPK) signaling, a key energy regulatory pathway, promoting inflammation and contributing to OA progression [86]. Additionally, lipopolysaccharides (LPSs) from Gram-negative bacterial cell walls can translocate into the bloodstream due to increased gut permeability. Once in circulation, LPSs activate Toll-like receptor 4 (TLR4) on immune cells, leading to the activation of the NF-κB signaling pathway, which induces the release of proinflammatory cytokines [87]. Moreover, other bacterial metabolites such as secondary bile acids and trimethylamine N-oxide (TMAO), derived from choline metabolism, exacerbate systemic inflammation and oxidative stress, further contributing to OA or RA pathogenesis [88,89,90]. TMAO, in particular, has been linked to increased cardiovascular risk and metabolic imbalances, which aggravate chronic inflammation and metabolic dysfunction, potentially accelerating OA progression. Collectively, these bacterial metabolites and signaling molecules—SCFAs, LPSs, secondary bile acids, and TMAO—impact metabolic pathways through interactions with GPCRs, TLR4, and the NF-κB pathway, leading to increased inflammation, cartilage degradation, and metabolic disturbances central to OA development.

Furthermore, gut dysbiosis can compromise gut barrier function, resulting in increased intestinal permeability and translocation of microbial products into systemic circulation. This phenomenon, often referred to as “leaky gut syndrome”, allows microbial-derived toxins and antigens to enter the bloodstream, where they can interact with immune cells and promote systemic inflammation [91,92]. These circulating microbial products can also infiltrate joint tissues, directly contributing to local inflammation and cartilage degradation. Additionally, gut dysbiosis may influence host immune responses and alter the balance of regulatory T cells and proinflammatory Th17 cells. Imbalances in these immune cell populations can exacerbate joint inflammation and tissue damage, further perpetuating OA pathology.

## 4. Interplay between Metabolic Dysfunction, GM, and OA

### 4.1. Crosstalk between Metabolic Signaling Pathways and GM in OA

Metabolic signaling pathways, including those involved in insulin resistance, adipokine secretion, and lipid metabolism, interact with the GM to influence OA pathophysiology. Dysregulated metabolic signaling pathways in conditions such as obesity and diabetes can impact GM composition and function, leading to gut dysbiosis [28]. In turn, alterations in GM composition can modulate metabolic processes, exacerbating metabolic dysfunction and promoting OA development [93]. Moreover, microbial-derived metabolites and signaling molecules can directly influence host metabolic signaling pathways, further shaping the metabolic milieu and contributing to OA pathogenesis.

### 4.2. Impact of Diet and Lifestyle Factors on Metabolic Health, GM, and OA Risk

Diet and lifestyle factors play a pivotal role in modulating metabolic health, GM composition, and OA risk. High-fat diets, excessive sugar consumption, and sedentary lifestyles are associated with metabolic dysfunction, gut dysbiosis, and increased OA risk [94]. Conversely, diets rich in fiber, fruits, and vegetables, along with regular physical activity, promote metabolic health, maintain GM diversity, and may reduce OA risk. These dietary and lifestyle interventions exert their effects on OA through multiple pathways, including the modulation of systemic inflammation, oxidative stress, and gut barrier integrity. Understanding the impact of diet and lifestyle factors on the interplay between metabolic dysfunction, the GM, and OA is essential for developing effective preventive and therapeutic strategies for OA management [95].

### 4.3. Gut Barrier Function and Its Role in Mediating the Relationship between Metabolic Dysfunction and OA

The gut barrier plays a critical role in maintaining intestinal homeostasis and preventing the translocation of harmful microbial products into systemic circulation [96,97]. The disruption of gut barrier function, commonly observed in metabolic disorders, can lead to increased intestinal permeability and leakage of microbial-derived toxins and antigens into circulation. These circulating toxins and antigens can trigger systemic inflammation, exacerbate metabolic dysfunction, and promote OA pathogenesis [98]. Moreover, impaired gut barrier function allows for the translocation of microbial products into joint tissues, where they can contribute to local inflammation and cartilage degradation [99]. Thus, preserving gut barrier integrity may represent a promising therapeutic target for mitigating the detrimental effects of metabolic dysfunction on OA development and progression [100].

## 5. Therapeutic Approaches

### 5.1. Lifestyle Interventions for Improving Metabolic Health and Modifying GM Composition

Lifestyle interventions, including dietary modifications and physical activity, represent cornerstone approaches for improving metabolic health and modifying OA composition in individuals with OA. Adopting a balanced diet rich in fiber, fruits, vegetables, and lean proteins while limiting the consumption of processed foods, saturated fats, and added sugars can promote metabolic health and support a diverse GM. Regular physical activity, including aerobic exercise and strength training, not only improves metabolic parameters but also exerts beneficial effects on GM composition. Lifestyle modifications may also contribute to weight management, reducing mechanical stress on weight-bearing joints and alleviating OA symptoms [94,101].

### 5.2. Pharmacological and Non-Pharmacological Interventions Targeting Metabolic Dysfunction and GM in OA

Both pharmacological and non-pharmacological interventions, such as diet and exercise, as well as pharmacologic treatments, including potential microbiome-targeted therapies, are crucial for comprehensive therapeutic management of musculoskeletal conditions like OP, sarcopenia, and OA [7,102]. Pharmacological interventions targeting metabolic dysfunction and GM composition offer promising avenues for OA management. Anti-diabetic medications such as metformin and thiazolidinediones have shown potential in mitigating OA progression by improving insulin sensitivity and reducing systemic inflammation [103]. Lipid-lowering agents such as statins and fibrates may also confer protective effects on OA by modulating lipid metabolism and inflammatory pathways. Furthermore, GM-targeted therapies, including antibiotics, probiotics, and FMT, are being explored for their potential to restore gut dysbiosis and alleviate OA symptoms. These pharmacological interventions hold promise for personalized approaches to OA management, targeting the underlying metabolic and microbial dysregulation driving OA pathophysiology. Moreover, given the high prevalence of OA and the significant therapeutic needs that remain unmet, it is essential to emphasize the role of rehabilitation therapies in OA management. A study on hip OA patients from 2018 to 2021 showed that combining drug therapy with rehabilitation, including physical and occupational therapy, led to better outcomes, particularly for those aged 41–50 [104]. This approach aligns with the broader strategy of targeting metabolic dysfunction and GM composition in OA management, offering a comprehensive and personalized treatment strategy that addresses both the physical and metabolic aspects of the condition.

### 5.3. Probiotics, Prebiotics, and Synbiotics as Potential Therapeutic Agents for OA Management

Probiotics, prebiotics, and synbiotics represent emerging therapeutic agents for OA management by modulating GM composition and promoting gut barrier integrity [95,105]. Probiotics, live microorganisms that confer health benefits when administered in adequate amounts, have shown potential in attenuating OA progression by restoring gut dysbiosis and reducing systemic inflammation [46,106]. Prebiotics, non-digestible dietary fibers that selectively promote the growth of beneficial gut bacteria, may enhance the efficacy of probiotics and support a healthy GM. Synbiotics, a combination of probiotics and prebiotics, offer synergistic effects on GM composition and metabolic health, providing a promising therapeutic approach for OA management. Future research is warranted to elucidate the specific strains and formulations of probiotics, prebiotics, and synbiotics that exert optimal effects on OA outcomes and to evaluate their long-term safety and efficacy in clinical settings.

## 6. Future Directions and Research Perspectives

As our understanding of the intricate relationship between metabolic dysfunction, the GM, and OA continues to evolve, future research efforts should focus on addressing key knowledge gaps and translating scientific discoveries into clinical applications. The research directions and perspectives outlined below offer insights into potential avenues for advancing our understanding of OA pathophysiology and developing innovative therapeutic strategies.

### 6.1. Identifying Novel Biomarkers for Predicting OA Risk and Treatment Response

Efforts to identify robust biomarkers for predicting OA risk and treatment response hold promise for early diagnosis, prognosis, and personalized therapeutic interventions. Integrating multiomics data, including genomics, transcriptomics, metabolomics, and microbiomics, may uncover novel biomarkers associated with OA susceptibility, severity, and progression [107]. Biomarker panels reflecting the interplay between metabolic dysfunction, GM composition, and OA pathophysiology may offer valuable insights into disease mechanisms and guide the development of targeted therapies [49,108]. Moreover, advances in imaging modalities, such as magnetic resonance imaging (MRI) and molecular imaging techniques, may enable the non-invasive detection of early OA changes and facilitate the monitoring of treatment responses in clinical settings.

### 6.2. Personalized Approaches for Targeting Metabolic Dysfunction and GM in OA

Moving toward personalized approaches for OA management is essential for optimizing treatment outcomes and improving patient care. Integrating clinical, genetic, and microbiomic data may enable the identification of patient subgroups with distinct metabolic and microbial profiles, allowing for tailored therapeutic interventions [109,110,111]. Precision medicine strategies, including pharmacogenomics and microbiota-based therapies, may help match patients with the most effective treatments based on their individual characteristics and disease phenotypes. Moreover, lifestyle interventions customized to individual preferences, socioeconomic factors, and cultural backgrounds may enhance patient adherence and long-term success in managing OA and associated comorbidities.

### 6.3. Integration of Multiomics Data to Better Understand the Complex Interplay between Metabolism, GM, and OA

Integrating multiomics data holds tremendous potential for unraveling the complex interplay between metabolism, the GM, and OA pathophysiology. Leveraging advanced bioinformatics tools and machine learning algorithms, researchers can analyze large-scale omics datasets to identify molecular signatures and biological pathways underlying OA development and progression. The integration of multiomics data may elucidate causal relationships between metabolic dysfunction, gut dysbiosis, and OA susceptibility, providing mechanistic insights into disease pathogenesis. Furthermore, systems biology approaches that integrate multiomics data with clinical phenotypes may facilitate the development of predictive models for OA risk stratification and treatment response prediction, paving the way for precision medicine approaches in OA management [82,112,113].

## 7. Conclusions

In this review, we synthesized the critical links between metabolic disorders, GM alterations, and OA, underscoring their shared risk factors and pathophysiological mechanisms. The increasing prevalence of obesity, diabetes, and sedentary lifestyles has amplified the global impact of OA. The GM’s role in metabolic regulation and immune response is pivotal, with dysbiosis contributing to both metabolic disorders and OA progression. Therapeutically, targeting metabolic dysfunction and GM composition presents a dual approach to OA management. Lifestyle modifications, pharmacotherapy, and microbiota-modulating therapies like probiotics, prebiotics, and synbiotics are promising for symptom relief and disease modification. Future research should focus on identifying OA biomarkers, precision medicine strategies, and the integration of multiomics to advance personalized OA treatment.

## Figures and Tables

**Figure 1 biomedicines-12-02182-f001:**
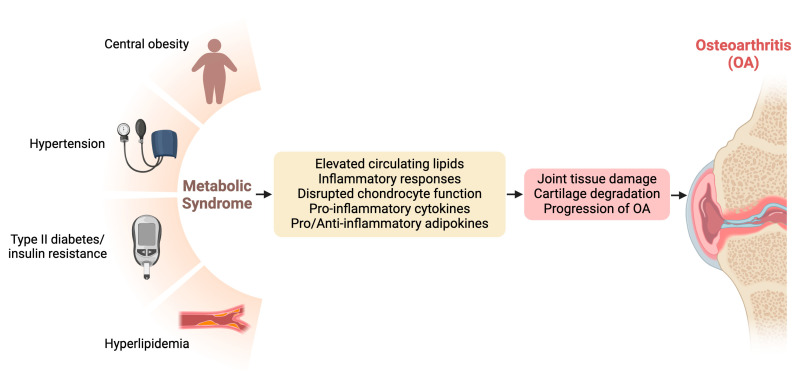
Interconnections between metabolic dysfunction and osteoarthritis (OA).

**Figure 2 biomedicines-12-02182-f002:**
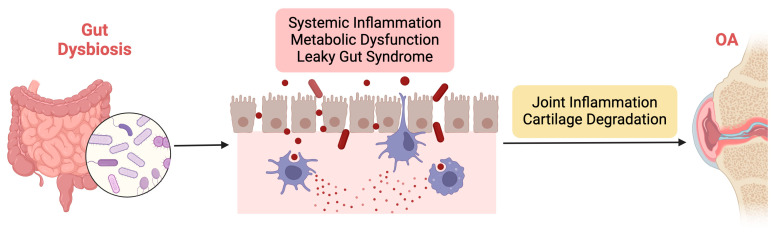
Illustration of the mechanisms underlying the influence of gut dysbiosis on OA pathophysiology.

**Table 1 biomedicines-12-02182-t001:** Recent studies on the relationship between GM and OA.

Main Findings	Species/Model	Intervention	Outcome	Refs.
The study highlighted the implications of GM for metabolic disorders and autoimmune diseases in canine and feline, including OA.	Canine, Feline	None	Overview of GM’s role in pet diseases	[47]
Alterations in the GM, including specific bacterial species, fungal species, and viral operational taxonomic units (vOTUs), differed between OA patients and healthy individuals.	Human	None	Revealed GM alterations in OA	[48]
The study identified differences in GM composition and functionality between OA patients and controls.	Human	None	Indicated the GM’s role in OA development	[49]
The study compared the GM structure between patients with knee OA and Kashin–Beck disease (KBD), identifying distinct microbial signatures associated with each condition.	Human	None	Revealed distinct GM in OA vs. KBD	[50]
The study revealed significant alterations in the GM composition and function in older female adults with OA compared to controls.	Human	None	Potential GM targets for OA treatment	[51]
A case–control study identified dysbiosis of the GM as a risk factor for OA in older female adults, suggesting potential GM targets for therapy.	Human	None	Identified GM dysbiosis as an OA risk factor	[52]
Gut fungal dysbiosis and an altered fungi–bacteria correlation network were associated with knee synovitis in a community-based study.	Human	None	Association between gut fungal microbiota and knee synovitis	[53]
The study found associations between GM-related metabolites and symptomatic hand OA in two independent cohorts.	Human	None	Association of GM metabolites with hand OA	[54]
The abundance of several GM species was associated with OA pathogenesis.	Human	None	Association between GM and OA	[25]
The abundance of Streptococcus species in the gut was associated with increased knee pain in a large population-based cohort study.	Human	None	Association between GM and joint pain	[55]
The study explored the use of GM alterations for inflammation and pain management in orthopedic conditions, including OA.	Human	Probiotics, Prebiotics	Management of inflammation and pain in OA	[56]
The study investigated the association between GM and elevated serum urate levels, finding significant alterations in GM functions related to tryptophan metabolism in individuals with symptomatic hand OA (SHOA).	Human	None	Association between GM functions and urate levels	[57]
The study used Mendelian randomization to establish a causal link between specific GM taxa and the development of knee OA, identifying Methanobacteriaceae, Desulfovibrionales, and Ruminiclostridium to be associated with knee OA.	Human	None	Causal relationship between GM taxa and OA	[24]
Live Lactobacillus acidophilus (LA-1) administration improved OA symptoms by modulating the GM and enhancing autophagy.	Mouse Model	Live LA-1	Improved pain threshold and joint health	[58]
Lactobacillus acidophilus treatment can reduce inflammatory knee joint pain and prevent further OA progression, possibly by modulating the GM.	Mouse Model	Lactobacillus acidophilus	Reduced pain and cartilage damage	[59]
Taxonomic changes in the GM are associated with cartilage damage in a mouse model of OA, independent of adiposity, high-fat diet, and joint injury.	Mouse Model	None	Association between GM and cartilage health	[60]
Gold nanoparticles (GNPs) exhibit anti-OA effects by modulating the microbiota–gut–joint axis, suggesting a novel therapeutic strategy for OA.	Mouse Model	GNPs	Alleviation of OA symptoms	[61]
Oral probiotics can ameliorate OA by modulating the GM.	Mouse Model	Probiotics	Alleviation of OA symptoms	[62]
The study investigated the protective potential of Bifidobacterium longum cell wall lipoproteins (Lpps) in preventing arthritis.	Mouse Model	Bifidobacterium longum Lpps	Prevention of arthritis	[63]
Oral administration of bovine milk-derived extracellular vesicles can attenuate cartilage degradation by modulating the GM in a mouse model of OA.	Mouse Model	Milk-derived extracellular vesicles	Attenuation of cartilage degradation	[64]
Fecal microbiota transplantation (FMT) from metabolically compromised human donors accelerates OA in mice.	Mouse Model	FMT	Acceleration of OA progression	[65]
Lactobacillus acidophilus (LA) treatment reduced OA-associated pain and cartilage disintegration and improved GM dysbiosis in a murine model.	Mouse Model	Lactobacillus acidophilus	Reduction in pain and cartilage damage, improved GM	[66]
GNPs alleviated OA by modulating the microbiota–gut–joint axis, increasing beneficial microbes and short-chain fatty acids, and reducing inflammation.	Mouse Model	GNPs	Alleviation of OA symptoms, modulation of GM	[67]
OA induced by destabilization of the medial meniscus is reduced in germ-free (GF) mice.	Mouse Model	None	Reduction in OA severity in GF mice	[68]
Antibiotic-induced GM dysbiosis alleviates the progression of OA in mice, potentially through reduced inflammation and improved bone parameters.	Mouse Model	Antibiotics	Alleviation of OA progression	[69]
GM depletion protects against bone loss and cartilage degradation in an OA and OP mouse model, modulating the GM composition.	Mouse Model	Antibiotics	Protection against bone loss and cartilage degradation	[70]
Moxibustion ameliorated OA by regulating GM and impacting the cAMP-related signaling pathway.	Mouse Model	Moxibustion	Amelioration of OA symptoms	[71]
OA susceptibility in mice is partially mediated by the GM, which is transferable via microbiome transplantation and associated with immunophenotype changes.	Mouse Model	Microbiome Transplantation	Reduction in OA severity with microbiome transplantation	[72]
The study provided evidence for a direct causal link between GM and bone diseases, including OA, mediated by neurophysiological states.	Human	None	A causal link between GM and bone diseases	[73]

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
