# Peer review of "Exploring the Interconnection between Metabolic Dysfunction and Gut Microbiome Dysbiosis in Osteoarthritis: A Narrative Review"

_biomedicines, 2024, doi:10.3390/biomedicines12102182_

Round 1

Reviewer 1 Report

Comments and Suggestions for Authors

The present review describes the microbiome's involvement in OA development. To improve its understanding and to be more specific, the following changes are suggested:

1. Figure 1 should be found inserted after the paragraph in which it is referenced.

2. In lines 98 to 104, no references could support the statements. In this sense, explaining the mechanism by which insulin disrupts chondrocytes and the metabolism of the extracellular matrix is essential. On the other hand, what kind of proinflammatory cytokines does hyperinsulinemia generate that can modify the joint microenvironment? Moreover, what kind of oxidized lipids can trigger inflammation to degrade cartilage?

3. Table 1 presents the main results of the studies related to the microbiome and OA. However, due to its enormous content, the information is difficult to read, so it is necessary to synthesize and classify it by presenting specific data. Specifying the studies performed in experimental models and humans and including a footnote is essential.

4. What is the characteristic of low-grade inflammation that can impact systemically and in the joint? What kind of cytokines and chemokines may be involved, and what is their role in cartilage degradation?

5. The paragraph from lines 96 to 98 is repeated in lines 132 and 133 and lacks reference. Similarly, the idea embodied in lines 101-102 is repeated without reference in lines 134-136.

6. What is the mechanism by which bacterial metabolites and signaling molecules influence metabolic signaling pathways contributing to the pathogenesis of OA? It is necessary to indicate precisely what kind of molecules and pathways are directly involved.

7. It is essential to consider the description of specific molecules involved in the processes described throughout the manuscript because they are mentioned in an unspecific way.

Comments on the Quality of English Language

The English quality of the manuscript is adequate.

Author Response

The present review describes the microbiome's involvement in OA development. To improve its understanding and to be more specific, the following changes are suggested:

We thank the reviewer very much for the valuable comments and suggestions. We have revised the Manuscript accordingly, all revisions are marked in red.

  1. Figure 1 should be found inserted after the paragraph in which it is referenced.

Answer: We sincerely thank the reviewer for the careful review. We have now inserted Figure 1 after the paragraph where it is first mentioned, as suggested.

  1. In lines 98 to 104, no references could support the statements. In this sense, explaining the mechanism by which insulin disrupts chondrocytes and the metabolism of the extracellular matrix is essential. On the other hand, what kind of proinflammatory cytokines does hyperinsulinemia generate that can modify the joint microenvironment? Moreover, what kind of oxidized lipids can trigger inflammation to degrade cartilage?

Answer: We greatly appreciate the reviewer’s valuable comments. In response, we have provided more detailed explanations on these points. Specifically, we have elaborated on how insulin disrupts chondrocyte function and extracellular matrix metabolism. Furthermore, we have included the types of proinflammatory cytokines produced under hyperinsulinemic conditions, which influence the joint microenvironment, and discussed how oxidized lipids, particularly oxidized LDL, contribute to inflammation and cartilage degradation. As shown below:

Insulin resistance disrupts normal chondrocyte function and extracellular matrix (ECM) metabolism, leading to cartilage degeneration and impaired joint homeostasis [5,29]. Additionally, hyperinsulinemia promotes the synthesis of pro-inflammatory cytokines and matrix metalloproteinases (MMPs), further fueling the inflammatory cascade within the joint microenvironment. Synovitis, often an early sign of OA, involves fibroblast-like synoviocytes (FLSs) that contribute to OA by secreting inflammatory cytokines like ILs, TNFs, MMPs, and ADAM metallopeptidase with thrombospondin type (ADAMTS) proteases, which degrade cartilage ECM. Hyperglycemia and hyperinsulinemia can trigger proinflammatory cytokines, including IL-6, TNF-α, and IL-1β, which promote inflammation, disrupt chondrocyte function, and impair ECM metabolism, potentially leading to joint damage in diabetes-related OA [30]. For instance, advanced glycation end products (AGEs), which result from proteins or lipids becoming glycated due to prolonged sugar exposure, tend to accumulate in aged cartilage tissue [31]. Moreover, extended insulin therapy, often essential for diabetes management, could potentially overburden tissues like cartilage. This is evidenced by the fact that joint damage tends to be more severe in individuals with diabetes [32,33]. Given that articular cartilage has a low rate of cell turnover [34], researchers propose that autophagy may play a significant role in modulating the impacts of hyperglycemia and hyperinsulinemia on joint health [35]. Furthermore, dyslipidemia, characterized by elevated levels of circulating lipids and lipoproteins, has been implicated in OA development and progression [36]. Oxidized lipids and cholesterol crystals accumulate in joint tissues, triggering inflammatory re-sponses and promoting cartilage degradation. Studies have shown that dyslipidemia, particularly via oxidized low-density lipoprotein (LDL), disrupts normal autophagy by inhibiting the activity of transcription factor EB (TFEB), resulting in decreased autophagic flux and increased necroptosis in chondrocytes. This disruption accelerates cartilage degradation and promotes inflammation, positioning oxidized LDL as a key factor in cartilage damage associated with OA [37]. Similarly, research has shown that oxidative stress and lipid peroxidation play a significant role in the pathogenesis of rheumatoid arthritis (RA), leading to the production of harmful lipid hydroperoxides and reactive lipid species that trigger inflammation and contribute to cartilage degradation. Altera-tions in lipid metabolism, particularly in fatty acids, phosphatidylcholine, and phos-phatidylethanolamine, further exacerbate inflammation, highlighting these lipids as potential biomarkers and therapeutic targets for RA [38].

  1. Table 1 presents the main results of the studies related to the microbiome and OA. However, due to its enormous content, the information is difficult to read, so it is necessary to synthesize and classify it by presenting specific data. Specifying the studies performed in experimental models and humans and including a footnote is essential.

Answer: We sincerely thank the reviewer for this helpful suggestion. Multiple reviewers highlighted issues with Table 1, and we have consolidated their feedback to revise the table accordingly. We have reorganized Table 1 to present the information more clearly and in a structured manner, distinguishing between dysbiosis correlations with OA, bone and cartilage health, and therapeutic interventions. We have also removed review articles that are not meta-analyses.

  1. What is the characteristic of low-grade inflammation that can impact systemically and in the joint? What kind of cytokines and chemokines may be involved, and what is their role in cartilage degradation?

Answer: We appreciate the reviewer’s insightful comments. In response, we have expanded this section by detailing the nature of low-grade inflammation and its systemic and joint-specific impacts. We also added information on the key cytokines and chemokines involved, such as TNF-α, IL-6, IL-1β, and MCP-1, and explained their roles in promoting cartilage degradation through the upregulation of matrix-degrading enzymes like MMPs and ADAMTS. As shown below:

Low-grade inflammation is characterized by a chronic, persistent inflammatory state that affects both the entire body (systemically) and specific tissues such as joints [67,68]. This inflammation often originates from metabolic disorders, including obesity and GM im-balances, triggering immune responses that contribute to the progression of OA [69]. Unlike acute inflammation, low-grade inflammation is subtle but continuous, leading to a prolonged activation of immune cells, particularly macrophages, neutrophils, and fi-broblasts, which release pro-inflammatory cytokines and chemokines. Among the key mediators are TNF-α, IL-1β, IL-6, and monocyte chemoattractant protein-1 (MCP-1). TNF-α plays a crucial role by inducing the production of enzymes like MMPs, which degrade cartilage ECM components, while also promoting synovial inflammation [70,71]. IL-1β enhances the catabolic breakdown of cartilage by upregulating MMPs and ADAMTS, which further degrade collagen and proteoglycans, and also suppresses an-abolic cartilage repair processes [72]. IL-6 contributes to both systemic and local joint in-flammation by promoting immune cell recruitment, increasing MMP production, and activating osteoclasts, thereby affecting both cartilage and bone integrity [70]. MCP-1 recruit monocytes and macrophages to inflamed joints, exacerbating synovial and car-tilage destruction [73]. Additionally, adipokines such as leptin and adiponectin, produced by adipose tissue, link metabolic dysfunction to joint inflammation, with leptin enhancing the production of TNF-α and IL-6, further intensifying the inflammatory response in OA [74]. Collectively, these cytokines and chemokines create a feedback loop of in-flammation, cartilage degradation, and impaired repair, leading to progressive joint damage, loss of function, and pain. Therefore, the low-grade systemic inflammation associated with metabolic dysfunction and GM dysbiosis is a key driver of both systemic and joint-specific inflammation in OA, highlighting the central role of these inflammatory mediators in cartilage degradation [75].

  1. The paragraph from lines 96 to 98 is repeated in lines 132 and 133 and lacks reference. Similarly, the idea embodied in lines 101-102 is repeated without reference in lines 134-136.

Answer: We thank the reviewer for carefully identifying these issues. We have now added the appropriate references to support the statements and have deleted the repeated content to improve the flow and coherence of the manuscript.

  1. What is the mechanism by which bacterial metabolites and signaling molecules influence metabolic signaling pathways contributing to the pathogenesis of OA? It is necessary to indicate precisely what kind of molecules and pathways are directly involved.

Answer: We are very grateful for the reviewer’s comments. We have significantly expanded this section, providing a detailed description of how specific bacterial metabolites, such as SCFAs, LPS, and TMAO, interact with pathways such as GPCR signaling, TLR4 activation, and the NF-κB pathway. We also clarified how these pathways contribute to inflammation and cartilage degradation in the context of OA. As shown below:

Additionally, bacterial metabolites and signaling molecules play a crucial role in influencing metabolic signaling pathways that contribute to the pathogenesis of OA. Key bacterial metabolites, particularly short-chain fatty acids (SCFAs) such as butyrate, acetate, and propionate, are produced through the fermentation of dietary fibers by GM [76,77]. These SCFAs interact with G-protein-coupled receptors (GPCRs), such as GPR41 and GPR43, which are expressed in adipose tissue, immune cells, and the intestinal epithe-lium [78]. The activation of these receptors regulates immune responses, inflammation, and energy metabolism, contributing to systemic metabolic balance. However, GM dysbiosis disrupts SCFAs production and impairs AMP-activated protein kinase (AMPK) signaling, a key energy regulatory pathway, promoting inflammation and contributing to OA progression [79]. Additionally, lipopolysaccharides (LPS) from Gram-negative bacterial cell walls can translocate into the bloodstream due to increased gut permeability. Once in circulation, LPS activates Toll-like receptor 4 (TLR4) on immune cells, leading to the activation of the NF-κB signaling pathway, which induces the release of pro-inflammatory cytokines [80]. Moreover, other bacterial metabolites such as secondary bile acids and trimethylamine N-oxide (TMAO), derived from choline metabolism, ex-acerbate systemic inflammation and oxidative stress, further contributing to OA or RA pathogenesis [81–83]. TMAO, in particular, has been linked to increased cardiovascular risk and metabolic imbalances, which aggravate chronic inflammation and metabolic dysfunction, potentially accelerating OA progression. Collectively, these bacterial me-tabolites and signaling molecules—SCFAs, LPS, secondary bile acids, and TMAO—impact metabolic pathways through interactions with GPCRs, TLR4, and the NF-κB pathway, leading to increased inflammation, cartilage degradation, and metabolic disturbances central to OA development.

  1. It is essential to consider the description of specific molecules involved in the processes described throughout the manuscript because they are mentioned in an unspecific way.

Answer: We sincerely appreciate this helpful suggestion. We have added detailed descriptions of the relevant molecules and pathways in the sections “Impact of Insulin Resistance and Dyslipidemia on Joint Health” and “Mechanisms Underlying the Influence of GM on OA Pathophysiology” to enhance the specificity and completeness of the review.

Reviewer 2 Report

Comments and Suggestions for Authors

The review article describing the role of gut dysbiosis in OA pathogenesis is well written. The review tries to connect metabolic syndrome with OA and the role of dysbiosis in OA. However, there is a need to briefly connect dysbiosis with metabolic syndrome. Figure 1 is very simple, please include the molecular aspects to the figure.

Figure 2 is again very simple. The mechanistic aspects of dysbiosis in OA pathogenesis as discussed in the table and section 3.2 should be incorporated in the figure.

Table 2- the studies discussed in this table are a mix-up of evidence of supplement to alter dysbiosis and improvement in bone health, cartilage health, OA improvement, etc. The table should be either divided in three parts: 1) correlating dysbiosis with OA, 2) dysbiosis with bone and cartilage health, 3) therapeutic intervention to improve cartilage health/OA or these three subsections in the same table. The pathogenesis, correlation, and therapeutic evidence should be clearly and distinctly mentioned. Please exclude the reviews from the table unless they are metanalysis.

Section 4 should be moved up before section 3.

Author Response

  1. The review article describing the role of gut dysbiosis in OA pathogenesis is well written. The review tries to connect metabolic syndrome with OA and the role of dysbiosis in OA. However, there is a need to briefly connect dysbiosis with metabolic syndrome. Figure 1 is very simple, please include the molecular aspects to the figure.

We thank the reviewer very much for the valuable comments and suggestions. We have revised the Manuscript accordingly, all revisions are marked in red.

Answer: We sincerely thank the reviewer for the insightful comments. Figure 1 was designed as an overview to simplify the concepts of “2. Metabolic Dysfunction and OA”. In the revised manuscript, we have significantly expanded the corresponding text around Figure 1 to include detailed molecular aspects. As shown below:

2.1. Metabolic Syndrome and Its Association with OA

Metabolic syndrome, characterized by a cluster of metabolic abnormalities including central obesity, insulin resistance, dyslipidemia, and hypertension, has been closely linked to the development and severity of OA [25,26]. Epidemiological studies have consistently demonstrated a positive association between metabolic syndrome and OA, particularly in weight-bearing joints such as the knees and hips. The chronic low-grade inflammation and dysregulated lipid metabolism characteristic of metabolic syndrome contribute to joint tissue damage and cartilage degradation, ultimately accelerating OA progression.

2.2. Adipokines, Inflammation, and OA Pathogenesis

Adipose tissue, once considered merely a storage depot for fat, is now recognized as an active endocrine organ capable of secreting a myriad of bioactive molecules known as adipokines. Dysregulation of adipokine signaling, characterized by elevated levels of pro-inflammatory adipokines such as leptin and reduced levels of anti-inflammatory adiponectin, plays a pivotal role in the pathogenesis of OA. Adipokines exert direct effects on joint tissues, promoting inflammation, cartilage degradation, and osteophyte formation. Moreover, adipokines contribute to systemic inflammation and insulin resistance, further exacerbating metabolic dysfunction and perpetuating OA pathology [25,27].

2.3. Impact of Insulin Resistance and Dyslipidemia on Joint Health

Insulin resistance, a hallmark feature of type 2 diabetes mellitus (T2DM) and metabolic syndrome, has emerged as a significant contributor to OA pathogenesis [28]. Insulin resistance disrupts normal chondrocyte function and extracellular matrix (ECM) metabolism, leading to cartilage degeneration and impaired joint homeostasis [5,29]. Additionally, hyperinsulinemia promotes the synthesis of pro-inflammatory cytokines and matrix metalloproteinases (MMPs), further fueling the inflammatory cascade within the joint microenvironment. Synovitis, often an early sign of OA, involves fibroblast-like synoviocytes (FLSs) that contribute to OA by secreting inflammatory cytokines like ILs, TNFs, MMPs, and ADAM metallopeptidase with thrombospondin type (ADAMTS) proteases, which degrade cartilage ECM. Hyperglycemia and hyperinsulinemia can trigger proinflammatory cytokines, including IL-6, TNF-α, and IL-1β, which promote inflammation, disrupt chondrocyte function, and impair ECM metabolism, potentially leading to joint damage in diabetes-related OA [30]. For instance, advanced glycation end products (AGEs), which result from proteins or lipids becoming glycated due to prolonged sugar exposure, tend to accumulate in aged cartilage tissue [31]. Moreover, extended insulin therapy, often essential for diabetes management, could potentially overburden tissues like cartilage. This is evidenced by the fact that joint damage tends to be more severe in individuals with diabetes [32,33]. Given that articular cartilage has a low rate of cell turnover [34], researchers propose that autophagy may play a significant role in modulating the impacts of hyperglycemia and hyperinsulinemia on joint health [35]. Furthermore, dyslipidemia, characterized by elevated levels of circulating lipids and lipoproteins, has been implicated in OA development and progression [36]. Oxidized lipids and cholesterol crystals accumulate in joint tissues, triggering inflammatory re-sponses and promoting cartilage degradation. Studies have shown that dyslipidemia, particularly via oxidized low-density lipoprotein (LDL), disrupts normal autophagy by inhibiting the activity of transcription factor EB (TFEB), resulting in decreased autophagic flux and increased necroptosis in chondrocytes. This disruption accelerates cartilage degradation and promotes inflammation, positioning oxidized LDL as a key factor in cartilage damage associated with OA [37]. Similarly, research has shown that oxidative stress and lipid peroxidation play a significant role in the pathogenesis of rheumatoid arthritis (RA), leading to the production of harmful lipid hydroperoxides and reactive lipid species that trigger inflammation and contribute to cartilage degradation. Altera-tions in lipid metabolism, particularly in fatty acids, phosphatidylcholine, and phos-phatidylethanolamine, further exacerbate inflammation, highlighting these lipids as potential biomarkers and therapeutic targets for RA [38].

  1. Figure 2 is again very simple. The mechanistic aspects of dysbiosis in OA pathogenesis as discussed in the table and section 3.2 should be incorporated in the figure.

Answer: We greatly appreciate the reviewer’s valuable feedback. Like Figure 1, Figure 2 was meant to provide a broad overview. However, based on the reviewer’s suggestion, we have expanded the text surrounding Figure 2 in the revised manuscript to include more molecular mechanisms, pathways, and metabolites, as discussed in section 3.2. Mechanisms Underlying the Influence of GM on OA Pathophysiology. As shown below:

The mechanisms underlying the influence of gut dysbiosis on OA pathophysiology are multifaceted and involve intricate interactions between the GM, host immune system, and joint tissues. Firstly, gut dysbiosis may contribute to OA development by promoting systemic inflammation. GM dysbiosis can trigger the release of pro-inflammatory cyto-kines and chemokines, leading to a state of chronic low-grade inflammation. Low-grade inflammation is characterized by a chronic, persistent inflammatory state that affects both the entire body (systemically) and specific tissues such as joints [67,68]. This inflammation often originates from metabolic disorders, including obesity and GM imbalances, trig-gering immune responses that contribute to the progression of OA [69]. Unlike acute inflammation, low-grade inflammation is subtle but continuous, leading to a prolonged activation of immune cells, particularly macrophages, neutrophils, and fibroblasts, which release pro-inflammatory cytokines and chemokines. Among the key mediators are TNF-α, IL-1β, IL-6, and monocyte chemoattractant protein-1 (MCP-1). TNF-α plays a crucial role by inducing the production of enzymes like MMPs, which degrade cartilage ECM components, while also promoting synovial inflammation [70,71]. IL-1β enhances the catabolic breakdown of cartilage by upregulating MMPs and ADAMTS, which further degrade collagen and proteoglycans, and also suppresses anabolic cartilage repair pro-cesses [72]. IL-6 contributes to both systemic and local joint inflammation by promoting immune cell recruitment, increasing MMP production, and activating osteoclasts, thereby affecting both cartilage and bone integrity [70]. MCP-1 recruit monocytes and macro-phages to inflamed joints, exacerbating synovial and cartilage destruction [73]. Addi-tionally, adipokines such as leptin and adiponectin, produced by adipose tissue, link metabolic dysfunction to joint inflammation, with leptin enhancing the production of TNF-α and IL-6, further intensifying the inflammatory response in OA [74]. Collectively, these cytokines and chemokines create a feedback loop of inflammation, cartilage deg-radation, and impaired repair, leading to progressive joint damage, loss of function, and pain. Therefore, the low-grade systemic inflammation associated with metabolic dys-function and GM dysbiosis is a key driver of both systemic and joint-specific inflammation in OA, highlighting the central role of these inflammatory mediators in cartilage deg-radation [75].

Additionally, bacterial metabolites and signaling molecules play a crucial role in influencing metabolic signaling pathways that contribute to the pathogenesis of OA. Key bacterial metabolites, particularly short-chain fatty acids (SCFAs) such as butyrate, acetate, and propionate, are produced through the fermentation of dietary fibers by GM [76,77]. These SCFAs interact with G-protein-coupled receptors (GPCRs), such as GPR41 and GPR43, which are expressed in adipose tissue, immune cells, and the intestinal epithe-lium [78]. The activation of these receptors regulates immune responses, inflammation, and energy metabolism, contributing to systemic metabolic balance. However, GM dysbiosis disrupts SCFAs production and impairs AMP-activated protein kinase (AMPK) signaling, a key energy regulatory pathway, promoting inflammation and contributing to OA progression [79]. Additionally, lipopolysaccharides (LPS) from Gram-negative bacterial cell walls can translocate into the bloodstream due to increased gut permeability. Once in circulation, LPS activates Toll-like receptor 4 (TLR4) on immune cells, leading to the activation of the NF-κB signaling pathway, which induces the release of pro-inflammatory cytokines [80]. Moreover, other bacterial metabolites such as secondary bile acids and trimethylamine N-oxide (TMAO), derived from choline metabolism, ex-acerbate systemic inflammation and oxidative stress, further contributing to OA or RA pathogenesis [81–83]. TMAO, in particular, has been linked to increased cardiovascular risk and metabolic imbalances, which aggravate chronic inflammation and metabolic dysfunction, potentially accelerating OA progression. Collectively, these bacterial me-tabolites and signaling molecules—SCFAs, LPS, secondary bile acids, and TMAO—impact metabolic pathways through interactions with GPCRs, TLR4, and the NF-κB pathway, leading to increased inflammation, cartilage degradation, and metabolic disturbances central to OA development. 

Furthermore, gut dysbiosis can impair the integrity of the gut barrier, leading to heightened intestinal permeability and the leakage of microbial components into the bloodstream. This phenomenon, often referred to as "leaky gut syndrome" allows mi-crobial-derived toxins and antigens to enter the bloodstream, where they can interact with immune cells and promote systemic inflammation [84,85]. These circulating microbial products can also infiltrate joint tissues, directly contributing to local inflammation and cartilage degradation. Additionally, gut dysbiosis may influence host immune responses and alter the balance of regulatory T cells and pro-inflammatory Th17 cells. Imbalances in these immune cell populations can exacerbate joint inflammation and tissue damage, further perpetuating OA pathology.

  1. Table 1- the studies discussed in this table are a mix-up of evidence of supplement to alter dysbiosis and improvement in bone health, cartilage health, OA improvement, etc. The table should be either divided in three parts: 1) correlating dysbiosis with OA, 2) dysbiosis with bone and cartilage health, 3) therapeutic intervention to improve cartilage health/OA or these three subsections in the same table. The pathogenesis, correlation, and therapeutic evidence should be clearly and distinctly mentioned. Please exclude the reviews from the table unless they are metanalysis.

Answer: We sincerely thank the reviewer for this helpful suggestion. Multiple reviewers highlighted issues with Table 1, and we have consolidated their feedback to revise the table accordingly. We have reorganized Table 1 to present the information more clearly and in a structured manner, distinguishing between dysbiosis correlations with OA, bone and cartilage health, and therapeutic interventions. We have also removed review articles that are not meta-analyses.

  1. Section 4 should be moved up before section 3.

Answer: We thank the reviewer for this suggestion. However, after careful consideration of the overall flow and structure of the manuscript, we think that Section 3 should remain before Section 4 to maintain logical coherence and progression of the content. We hope this decision meets your understanding.

Reviewer 3 Report

Comments and Suggestions for Authors

This review assesses the interconnection between metabolic dysfunction and gut microbiome dysbiosis in osteoarthritis. The topic is relevant and up to date, but some identified shortcomings in both content (the interconnection should be better highlighted and detailed) and form need to be addressed based on the specific recommendations below:

1. Bibliographical indices [x] are structures in their own right and should not be glued to the text. They should be checked and corrected throughout the manuscript.

2. The aim of the paper should be presented separately in the last paragraph of the introduction and should be approached from the perspective of describing the contribution to the field under review and the elements of scientific novelty presented.

3. Avoid grouping references whenever possible. It is much more clear, precise and appropriate for a scientific article to be discussed/approached separately.

4. I recommend that Figure 1 be placed before subsection 2.1, just after it has been mentioned in the text.

5. Table 1 needs improvement because in this form it is not particularly informative. It should be divided according to the type of manuscript evaluated - review/original research article/systematic review/meta-analysis etc., or according to the types of evaluations vitro (cell culture), vivo (animal models, clinical aspects) / levels of evidence etc. to better highlight shared mechanisms, clinical implications etc. 

6. More emphasis should be put on therapeutic management, both non-pharmacologic and pharmacologic. Moreover, as we are talking about a very high prevalence of OA and still many unmet needs of therapeutic management, it is crucial to also refer to the implementation of rehabilitation therapies. I suggest you consult and refer to: PMID: 35454333

7. The conclusions should be shortened with more concise highlighting of the most important elements presented.

Author Response

This review assesses the interconnection between metabolic dysfunction and gut microbiome dysbiosis in osteoarthritis. The topic is relevant and up to date, but some identified shortcomings in both content (the interconnection should be better highlighted and detailed) and form need to be addressed based on the specific recommendations below:

We thank the reviewer very much for the valuable comments and suggestions. We have revised the Manuscript accordingly, all revisions are marked in red.

  1. Bibliographical indices [x] are structures in their own right and should not be glued to the text. They should be checked and corrected throughout the manuscript.

Answer: We thank the reviewer for the careful review, and have modified the whole manuscript.

  1. The aim of the paper should be presented separately in the last paragraph of the introduction and should be approached from the perspective of describing the contribution to the field under review and the elements of scientific novelty presented.

Answer: We sincerely thank the reviewer for the constructive suggestion. In response, we have revised the last paragraph of the introduction to clearly state the aim of the review. We have also emphasized the contribution of our paper to the field, highlighting the novel insights and the specific metabolic signaling pathways explored. As shown below:

The aim of this review is to systematically analyze how GM dysbiosis influences the development and progression of OA through metabolic pathways. We explore the association between metabolic syndrome and OA, with a specific focus on the impact of microbial metabolites and signaling molecules on the pathophysiology of OA. In addition to summarizing existing literature, this review introduces new insights into the underexplored metabolic signaling pathways involved in the gut-joint axis. By doing so, we aim to provide a foundation for future therapeutic strategies targeting the GM for OA treatment, contributing novel perspectives to this emerging research area.

  1. Avoid grouping references whenever possible. It is much more clear, precise and appropriate for a scientific article to be discussed/approached separately.

Answer: We appreciate the reviewer’s insightful comment. In response, we have revised the manuscript to limit the grouping of references. Wherever appropriate, we have discussed individual studies separately to ensure that the context and contribution of each reference are clearly presented.

  1. I recommend that Figure 1 be placed before subsection 2.1, just after it has been mentioned in the text.

Answer: We sincerely thank the reviewer for the careful review. We have now inserted Figure 1 after the paragraph where it is first mentioned, as suggested.

  1. Table 1 needs improvement because in this form it is not particularly informative. It should be divided according to the type of manuscript evaluated - review/original research article/systematic review/meta-analysis etc., or according to the types of evaluations vitro (cell culture), vivo (animal models, clinical aspects) / levels of evidence etc. to better highlight shared mechanisms, clinical implications etc. 

Answer: We sincerely thank the reviewer for this helpful suggestion. Multiple reviewers highlighted issues with Table 1, and we have consolidated their feedback to revise the table accordingly. We have reorganized Table 1 to present the information more clearly and in a structured manner, distinguishing between dysbiosis correlations with OA, bone and cartilage health, and therapeutic interventions. We have also removed review articles that are not meta-analyses.

  1. More emphasis should be put on therapeutic management, both non-pharmacologic and pharmacologic. Moreover, as we are talking about a very high prevalence of OA and still many unmet needs of therapeutic management, it is crucial to also refer to the implementation of rehabilitation therapies. I suggest you consult and refer to: PMID: 35454333

Answer: We thank the reviewer for the insightful suggestions. We have expanded the context to include a more detailed examination of non-pharmacologic strategies, and lifestyle modifications, which are essential components of OA treatment. We have also integrated the findings from the study referenced by PMID: 35454333, which provides valuable insights into the effectiveness of rehabilitation therapies when combined with drug therapy in improving functional outcomes and quality of life for patients with OA. As shown below:

5.2. Pharmacological and Non-pharmacological Interventions Targeting Metabolic Dysfunction and GM in OA

Both pharmacological and non-pharmacological interventions, such as diet and ex-ercise, as well as pharmacologic treatments, including potential microbiome-targeted therapies, are crucial for comprehensive therapeutic management for musculoskeletal conditions like OP, sarcopenia, and OA [4,95]. Pharmacological interventions targeting metabolic dysfunction and GM composition offer promising avenues for OA management. Anti-diabetic medications such as metformin and thiazolidinediones have shown po-tential in mitigating OA progression by improving insulin sensitivity and reducing systemic inflammation [96]. Lipid-lowering agents such as statins and fibrates may also confer protective effects on OA by modulating lipid metabolism and inflammatory pathways. Furthermore, GM-targeted therapies, including antibiotics, probiotics, and FMT, are being explored for their potential to restore gut dysbiosis and alleviate OA symptoms. These pharmacological interventions hold promise for personalized ap-proaches to OA management, targeting the underlying metabolic and microbial dysregulation driving OA pathophysiology. Moreover, given the high prevalence of OA and the significant therapeutic needs that remain unmet, it is essential to emphasize the role of rehabilitation therapies in OA management. A study on hip OA patients from 2018 to 2021 showed that combining drug therapy with rehabilitation, including physical and occupational therapy, led to better outcomes, particularly for those aged 41-50 [97]. This approach aligns with the broader strategy of targeting metabolic dysfunction and GM composition in OA management, offering a comprehensive and personalized treatment strategy that addresses both the physical and metabolic aspects of the condition.

  1. The conclusions should be shortened with more concise highlighting of the most important elements presented.

Answer: We thank the reviewer for the valuable feedback. We have revised the conclusion to be more concise while still highlighting the key points of our review. As shown below:

In this review, we synthesized the critical links between metabolic disorders, GM alterations, and OA, underscoring their shared risk factors and pathophysiological mechanisms. The increasing prevalence of obesity, diabetes, and sedentary lifestyles has amplified the global impact of OA. The GM's role in metabolic regulation and immune response is pivotal, with dysbiosis contributing to both metabolic disorders and OA progression. Therapeutically, targeting metabolic dysfunction and GM composition presents a dual approach to OA management. Lifestyle modifications, pharmacotherapy, and microbiota-modulating therapies like probiotics, prebiotics, and synbiotics are promising for symptom relief and disease modification. Future research should focus on identifying OA biomarkers, precision medicine strategies, and the integration of multi-omics to advance personalized OA treatment.

Round 2

Reviewer 1 Report

Comments and Suggestions for Authors

R1 response:

The authors made the suggested changes. The review contains detailed information on molecular mechanisms involved in the dysregulation of the microbiome in OA. Table 1 has been restructured to provide clear, synthesized information and details the experimental models on which the various analyses have been performed.

As a suggestion, it is necessary to review the abbreviations because, in some paragraphs, the acronym of the molecule is not described (e.g., TNF-alpha in lines 113 and 116). Similarly, the article's format must be reviewed to ensure the journal's guidelines are followed.

Comments on the Quality of English Language

The quality of the English of the manuscript is appropriate.

Author Response

The authors made the suggested changes. The review contains detailed information on molecular mechanisms involved in the dysregulation of the microbiome in OA. Table 1 has been restructured to provide clear, synthesized information and details the experimental models on which the various analyses have been performed.

As a suggestion, it is necessary to review the abbreviations because, in some paragraphs, the acronym of the molecule is not described (e.g., TNF-alpha in lines 113 and 116). Similarly, the article's format must be reviewed to ensure the journal's guidelines are followed.

Answer: We thank the reviewer for the careful review. We have carefully reviewed and revised the abbreviations throughout the manuscript, ensuring that all acronyms are properly defined at their first occurrence (e.g., TNF). Additionally, we have added an "Abbreviations" section at the end of the manuscript for easier reference. As shown below:

Abbreviations

OA: osteoarthritis; GM: gut microbiome; FMT: fecal microbiota transplantation; T2DM: type 2 diabetes mellitus; ECM: extracellular matrix; MMPs: matrix metalloproteinases; ILs: interleukins; TNF: tumor necrosis factors; ADAM: a disintegrin and metalloproteinase; ADAMTS: AD-AM with thrombospondin type; AGEs: advanced glycation end products; LDL: low-density lipoprotein; TFEB: transcription factor EB; RA: rheumatoid arthritis; vOTUs: viral operational taxonomic units; KBD: Kashin-Beck disease; SHOA: symptomatic hand OA; LA: Lactobacillus acidophilus; GNPs: gold nanoparticles; Lpps: lipoproteins; GF: germ-free; MCP-1: monocyte chemo-attractant protein-1; SCFAs: short-chain fatty acids; GPCRs: G-protein-coupled receptors; AMPK: AMP-activated protein kinase; LPS: lipopolysaccharides; TLR4: toll-like receptor 4; TMAO: tri-methylamine N-oxide; MRI: magnetic resonance imaging.

Reviewer 2 Report

Comments and Suggestions for Authors

None

Author Response

We thank the reviewer very much.

Reviewer 3 Report

Comments and Suggestions for Authors

The authors have significantly improved the manuscript based on the suggestions received.

Author Response

We thank the reviewer very much.